# VSCD: Video-based Scene Change Detection in Unaligned Scenes

**Jiae Yoon** [1]   **Ue-Hwan Kim** [1 2]

## Abstract

Detecting what has changed in an environment is essential for long-term autonomy, yet most change detection settings assume fixed viewpoints, mild misalignment, or only a few changed objects. We introduce Video-based Scene Change Detection (VSCD), which predicts a pixel-wise change mask for each query frame, given a reference and a query RGB video of the same indoor space recorded at different times under unconstrained camera motion. The two videos are not temporally synchronized, and many object instances may appear or disappear. To study this setting, we build a large-scale benchmark with over 1.1 million frames annotated with pixel-accurate change masks, together with a real-world test set for evaluating transfer beyond simulation. We propose a query-centric multi-reference model that learns temporal matching implicitly from change-mask supervision, aligns candidate reference features to the query via local patch correspondence, and fuses per-candidate change features using frame-level and patch-level confidence before decoding a high-resolution mask once per frame. Our approach achieves state-of-the-art performance against strong image- and video-based baselines, and we validate its real-world impact by deploying it on a mobile robot for two downstream applications—visual surveillance and object incremental learning.

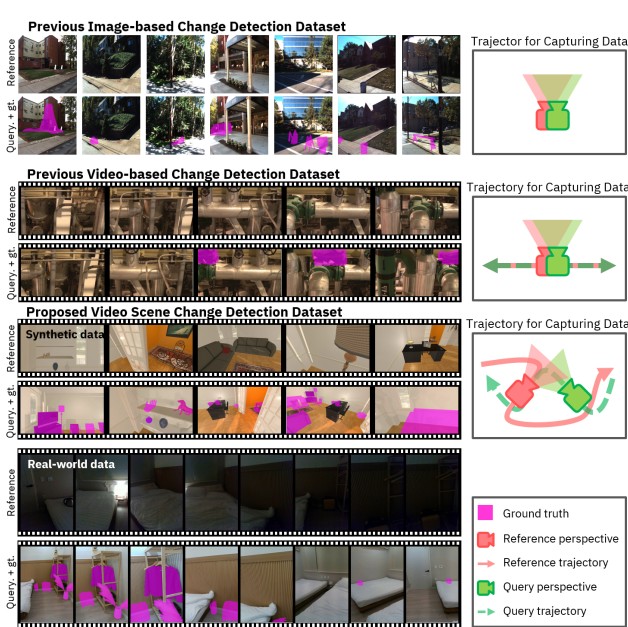

*Figure 1.* **Comparison between existing change detection datasets and our VSCD benchmark.** Unlike prior image-based and video-based datasets that assume similar viewpoints or trajectories, VSCD features unconstrained camera motion, which leads to strong misalignment between reference and query videos.

## 1. Introduction

Embodied agents that operate autonomously in real environments face a crucial challenge: the world is non-stationary and ever-changing. In dynamic surroundings, humans do not process all visual information uniformly. Instead, the vision system is particularly sensitive to *changes* in the environment, such as the appearance or motion of objects, and indeed detects such changes reliably (Yantis & Jonides, 1984; Cole et al., 2004): change signals function as a key cue to summarize complex scenes and to prioritize behaviorally relevant information (Rensink, 2002; Czigler & Kojouharova, 2022). Thus, the importance of change is no less for embodied agents in the physical world (Luo et al., 2018; Driess et al., 2023)—not only recognizing what is present at a single time but also accurately detecting what has changed over time.

This requirement naturally leads to the change detection task, whose goal is to identify semantically meaningful

---

[1]Department of AI Convergence, Gwangju Institute of Science and Technology, Gwangju, Republic of Korea [2]GIST InnoCORE AI-Nano Convergence Institute for Early Detection of Neurodegenerative Diseases, Gwangju Institute of Science and Technology, 61005 Gwangju, Republic of Korea. Correspondence to: Ue-Hwan Kim <uehwan@gist.ac.kr>.

*Proceedings of the 43$^{rd}$ International Conference on Machine Learning*, Seoul, South Korea. PMLR 306, 2026. Copyright 2026 by the author(s).

*Table 1.* **Comparison of change detection task formulations.** We contrast existing benchmarks in terms of input modality, motion assumptions, alignment difficulty, and change complexity.

| Task | RSCD | SCD | AOD | VSCD (Ours) |
|---|---|---|---|---|
| **Input modality** | 2 images | 2 images | 2 videos | 2 videos |
| **Camera mobility** | Static | Static | Limited | Unconstrained |
| **Viewpoint variation** | Low | Medium | Low | High |
| **Trajectory mismatch** | None | None | Limited | High |
| **Temporal reasoning** | X | X | △ | O |
| **Change type** | Land-cover / Structure | Object appear / disappear | Object appear | Object appear / disappear |
| **Scale of changes** | Large regions | Few objects | Single / few objects | Many objects |

changes between observations of the same environment at different times. However, conventional change detection methods fail in the presence of conditions that naturally arise in real-world settings: unconstrained camera motion, strong viewpoint misalignment, and multiple simultaneous changes. Specifically, image-based methods only guarantee the reported performance when viewpoint variation is modest (Lu et al., 2004; Sakurada & Okatani, 2015), and video-based approaches assume nearly identical camera trajectories and conduct frame-wise change detection (Padilla et al., 2023; Tavares et al., 2025). This observation reveals an essential gap between current methodology and the requirements of long-term autonomy.

To bridge the gap, we formulate the task from a fundamentally different angle. Our formulation aims to detect changes between two unconstrained RGB videos of the same indoor environment observed at different times as a mobile agent independently traverses the scene. We define change at the level of objects and treat variations affecting only image appearance—such as illumination, shadows, or reflections—as nuisance factors rather than changes. Major characteristics of our Video-based Scene Change Detection (VSCD) include *unconstrained camera motion*, *severe viewpoint variation*, *strong cross-view misalignment*, *occlusions*, and *multiple simultaneous changes*. To systematically study this setting, we construct a large-scale benchmark comprising more than 1.1 million frames with high-quality pixel-wise change masks, along with a real-world test set to evaluate generalization.

Furthermore, our central reconceptualization is from *independent frame-wise comparisons* to *sequence-level alignment that jointly leverages temporal continuity and multiview geometry*. While individual frames may be severely misaligned, sequences encode sufficient temporal and spatial regularity to enable robust reasoning. This perspective motivates a three-stage architecture: (1) frame-level alignment (temporal consistency) across reference and query sequences without trajectory alignment, (2) patch-level correspondence (geometric compensation) to learn multiview geometry without explicit pose supervision, and (3) confidence-weighted fusion (robustness) to suppress am-

biguous matches.

In summary, we make the following contributions:

- **Problem Formulation**: We introduce the first VSCD task that captures realistic scenarios essential for long-term autonomous operation.

- **Large-Scale Benchmark**: We create and release a curated dataset combining over 1.1 million frames with real-world examples for rigorous evaluation.

- **Sequence-Aware Architecture**: Our model exploits temporal structure without explicit supervision, outperforming strong baselines.

- **Real-World Validation**: Practical deployment on Stretch demonstrates the method's applicability beyond standard benchmarks.

- **Open Source**: To facilitate research on VSCD for mobile agents, we publicly release the full implementation of our method at `https://github.com/AutoCompSysLab/VSCD`.

## 2. Related Work

### 2.1. Change Detection

Change detection aims to identify differences in observations of the same environment at different times. We categorize methods into image-based and video-based approaches depending on input modality. Table 1 compares existing change detection tasks.

#### 2.1.1. CHANGE DETECTION IN IMAGE

**Remote sensing change detection (RSCD)** employs satellite or aerial imagery—whose viewpoint, scale, and texture statistics differ markedly from standard indoor or street-view images—to capture large-scale environmental changes such as land-cover or building transitions (Gupta et al., 2019; Chen & Shi, 2020; Shi et al., 2021). Early works have relied on patch-based classifiers (Chen et al., 2019; 2020) or CNN

variants (Daudt et al., 2018; Fang et al., 2021), while recent methods adopt Transformer-based models (Chen et al., 2021a). However, RSCD operates on well-aligned image pairs and struggles in dynamic environments with significant viewpoint changes or occlusions.

**Scene change detection (SCD)** localizes object-level changes in street-view or indoor images from different times (Sakurada & Okatani, 2015; Alcantarilla et al., 2018; Sakurada et al., 2020; Park et al., 2021). Representative approaches include a Siamese architecture (Daudt et al., 2018), feature similarity via correlation layers (Sakurada et al., 2020), a temporal attention module (Chen et al., 2021b), tuned segmentation networks (Wang et al., 2023), and adapted foundation models (Cho et al., 2025; Kim & Kim, 2025). While SCD targets object-level changes similar to VSCD and handles slight misalignment, it operates on isolated image pairs and cannot exploit temporal structure or multiple viewpoints, which are central to VSCD.

### 2.1.2. CHANGE DETECTION IN VIDEO

**Abandoned object detection (AOD)** detects abandoned objects by comparing a query video and an anomaly-free reference video in surveillance settings (da Silva et al., 2014). Various approaches for the AOD task have emerged, such as modelling the reference video as a union of low-dimensional subspaces (Thomaz et al., 2017), stacking reference and query frames into a single matrix and applying a domain transformation for alignment (Jardim et al., 2019), low-complexity dissimilarity module to compute frame-wise differences (Padilla et al., 2023), and combining multiple CNN-based feature extractors with a tree-based classifier (Tavares et al., 2025). Although AOD extends change detection to videos, its assumption of constrained camera motion—i.e., the reference and query videos usually follow the same or opposite trajectories—and its small number of target objects—often limited to a single instance—limit its generality compared to VSCD.

### 2.2. Video Comparison

Video comparison takes two videos as input and determines whether, and to what extent, one has been copied or is a near-duplicate of the other.

### 2.2.1. VIDEO COPY DETECTION

Video copy detection has focused on determining whether a query video contains content copied from a reference video, even under temporal, geometric, or photometric transformations (Lian et al., 2010). For example, DML-NDVR (Kordopatis-Zilos et al., 2017) samples frames and forms a single video descriptor, which is then compared; CR-UML (Cheng et al., 2021) performs k-nearest-neighbor retrieval and iteratively refines the embedding in an unsupervised

manner; and a recent method dynamically selects representative frames and encodes a highly compressed video descriptor to reduce computational and storage costs (Fojcik et al., 2025).

### 2.2.2. VIDEO COPY LOCALIZATION

Recent work extends beyond binary detection to localize exact copied segments between two videos (Han et al., 2021). For instance, VSAL (Han et al., 2021) constructs a frame-wise spatial similarity map to recover the optimal alignment path for partial copies; TransVCL (He et al., 2023) uses a dual-softmax operation to obtain a differentiable similarity map to predict the temporal boundaries of copied segments; the Similarity Alignment Model (Liu et al., 2023) adopts a high-resolution network to refine similarity maps to highlight copied regions; and RTR (Lu et al., 2024) augments each frame with a regional token to encode local information to deal with a small spatial region.

Notably, video copy localization identifies copied segments at the frame or temporal level and is not designed for understanding pixel-wise or object-level changes. Inspired by frame-wise similarity maps for localization, we adopt similar principles but extend them to pixel-wise semantic change detection.

## 3. VSCD Task

### 3.1. Problem Formulation

We consider the problem of detecting pixel-wise object-level changes between two RGB videos of the same physical environment, captured at two different time points under unconstrained camera motion. We do not assume any known geometric calibration, temporal synchronization, or frame-to-frame alignment between the two videos. Within each video, the scene state is fixed, and object-level changes occur only between the two recordings; temporal continuity is used as a structural prior for cross-video matching.

Let $H$ and $W$ denote the height and width of video frames, and define the pixel domain as $\Omega = \{1,\ldots,H\} \times \{1,\ldots,W\}$. We are given a reference video $V_r \in \mathbb{R}^{T_r \times H \times W \times 3}$ and a query video $V_q \in \mathbb{R}^{T_q \times H \times W \times 3}$, where $T_r$ and $T_q$ are the numbers of frames in each video. We write $V_r = \{F_r^t\}_{t=1}^{T_r}, V_q = \{F_q^t\}_{t=1}^{T_q}$, with each frame $F_r^t, F_q^t \in \mathbb{R}^{H \times W \times 3}$.

Let $O$ denote the universe of object occupancies, where each occupancy is an object instance together with its physical location. The environment at the reference and query times induces finite sets $O_r, O_q \subset O$. We define appearances and disappearances as $O_{\text{app}} = O_q \setminus O_r$ and $O_{\text{dis}} = O_r \setminus O_q$, and the set of object-level changes as the symmetric difference $O_\Delta = O_{\text{app}} \cup O_{\text{dis}}$. Under this definition, relocating an object is also treated as a change (a disappearance at the original

*Table 2.* **Cross-dataset scale and diversity.** *Video units* denotes the number of capture units (pairs/sequences/videos). *Views* counts labeled frames/images, and *Locs* counts distinct environments.

| Dataset | Video units | Views | Locs |
|---|---|---|---|
| VDAO (da Silva et al., 2014) | 66 videos | 0.71M | 1 |
| VL-CMU-CD (Alcantarilla et al., 2018) | – | 1,362 | 1 |
| ChangeSim (Park et al., 2021) | 80 seq. | 130k | 10 |
| VSCD-Syn (Ours) | 1,090 pairs | 1.13M | 218 |
| VSCD-Real (Ours) | 8 pairs | 17.0k | 8 |

location and an appearance at the new location). Variations that affect only appearance (e.g., illumination, shadows, reflections) are regarded as nuisance factors and are not considered object-level changes.

For a query frame $F_q^t$, each pixel $(i, j) \in \Omega$ corresponds to a visible physical location (up to occlusion), and we state it *changed* if the associated object occupancy belongs to $O_\Delta$, i.e., it is present at exactly one of the two time points. Then, the ground-truth change mask for the $t$-th query frame is a binary image $M^t \in \{0, 1\}^{H \times W}$, whose value at pixel $(i, j)$ is

$$M_{i,j}^t = \begin{cases} 1 & \text{if pixel } (i, j) \text{ in } F_q^t \text{ is a physical location} \\ & \text{associated with an object instance in } O_\Delta, \\ 0 & \text{otherwise.} \end{cases} \tag{1}$$

$M_{i,j}^t = 1$ accounts for viewpoint changes and occlusions between the two recordings. Stacking all frames yields the ground-truth change mask sequence

$$M = \{M^t\}_{t=1}^{T_q} \in \{0, 1\}^{T_q \times H \times W}. \tag{2}$$

A VSCD method is any mapping $g$ that produces a per-pixel change score for each query frame and generates $\hat{M}$:

$$\hat{Z} = g(V_r, V_q) \in \mathbb{R}^{T_q \times H \times W}, \qquad \hat{M} = \mathbb{I}\big[\rho(\hat{Z}) > \tau\big], \tag{3}$$

where $\rho(\cdot)$ is an element-wise score-to-confidence mapping and $\tau$ is a decision threshold. In this work, we instantiate $g$ with a parameterized model $f_\theta$ and learn $\theta$ from change-mask supervision.

### 3.2. Benchmark Dataset

We build a dedicated benchmark for the VSCD task that combines a large-scale synthetic dataset with a carefully annotated real-world test set. Each example consists of a reference video, a query video recorded in the same physical environment at a different time, and a pixel-wise change-mask sequence for the query video. The benchmark is designed to stress the key challenges of VSCD, namely free agent motion, strong cross-view misalignment, and many simultaneous object-level changes. Table 2 summarizes the scale and environmental diversity of our benchmark in comparison to representative change detection datasets.

#### 3.2.1. SYNTHETIC DATA

We generate synthetic VSCD data in two simulation stacks (AI2-THOR (Kolve et al., 2017b) and Unreal Engine) to introduce rendering diversity while preserving the same VSCD input format. Within each layout, we form directed scene pairs according to a fixed cycle (e.g., $0 \to 1, 1 \to 2, \ldots, 4 \to 0$), so that every scene appears once as the query. For each environment, we substantially vary object instance configurations across time and capture reference and query videos along distinct, manually designed trajectories to induce strong cross-view misalignment and occlusions. To better reflect real-world conditions, some scene pairs are rendered under different illumination due to changes in daylight and indoor lighting, which can significantly alter color, brightness, and shadows, thereby making appearance-only cues less reliable. Pixel-accurate query-aligned masks are obtained by replaying the query trajectory and rendering only the changed occupancies $O_\Delta$.

#### 3.2.2. REAL-WORLD DATA

To measure sim-to-real transfer under real sensing artifacts, we collect eight indoor reference–query video pairs with notable viewpoint mismatch and no temporal synchronization (three robot-captured and five handheld). Environments undergo object-level additions, removals, and relocations between recordings, and the real set is used exclusively for evaluation while training is performed on the synthetic split.

## 4. Methodology

### 4.1. Overview

We propose VSCDNet, a query-centric multi-reference model for VSCD (Fig. 2). Given two unaligned videos, VSCDNet learns to associate each query keyframe with multiple reference keyframes and to predict a pixel-wise change mask. Our design follows three principles: (i) **Frame-level alignment** that enforces temporal consistency across sequences without trajectory synchronization, (ii) **Patch-level correspondence** that performs local geometric compensation without pose supervision, and (iii) **Confidence-weighted fusion** that suppresses ambiguous matches before high-resolution decoding.

### 4.2. Stage 1: Frame-level Alignment and Reference Candidate Construction

**Keyframe encoding and frame representation.** We uniformly sample $T_{\text{key}}$ keyframes from each video and encode each frame using a frozen pretrained ViT image encoder (SAM-ViT (Kirillov et al., 2023)). The encoder outputs patch tokens on a fixed grid, which serve as the shared representation for both temporal matching and local correspondence. To obtain a compact frame descriptor while

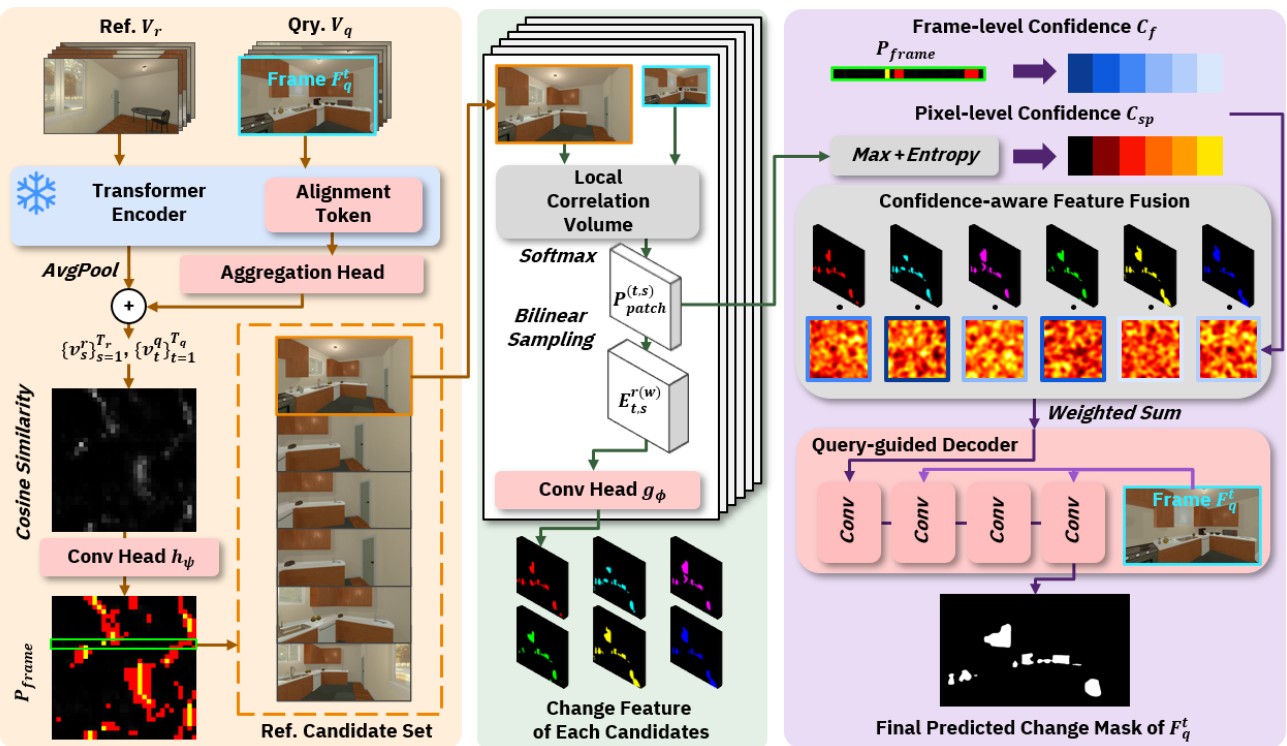

*Figure 2.* **The architecture of the proposed VSCDNet.** The model consists of three stages: frame-level alignment to select reference candidates, patch-level correspondence for viewpoint compensation, and confidence-aware fusion followed by query-guided decoding to predict change masks.

keeping the encoder frozen, we use a learnable alignment token updated by a lightweight attention-based aggregation head attending to patch tokens, and combine it with mean pooling:

$$v_t = \text{AvgPool}(E_t) + \text{AT}(E_t), \tag{4}$$

where $E_t$ denotes patch tokens of the frame and $\text{AT}(\cdot)$ outputs the updated alignment-token embedding. Applying the same construction to $E_t^q$ and $E_s^r$ yields the query/reference frame vectors $v_t^q$ and $v_s^r$, respectively.

**Frame similarity and soft matching.** We compute a cosine similarity grid $S \in \mathbb{R}^{T_{\text{key}} \times T_{\text{key}}}$ with $S_{t,s} = \cos(v_t^q, v_s^r)$. A shallow residual convolutional head $h_\psi : \mathbb{R}^{T_{\text{key}} \times T_{\text{key}}} \to \mathbb{R}^{T_{\text{key}} \times T_{\text{key}}}$ refines the grid, and we define the refined alignment logits by element-wise addition $A = S + h_\psi(S)$. A row-wise softmax with temperature $\tau_f$ yields a soft matching distribution:

$$P_{\text{frame}}(t, s) = \frac{\exp(A_{t,s}/\tau_f)}{\sum_{s'} \exp(A_{t,s'}/\tau_f)}. \tag{5}$$

We use $P_{\text{frame}}(t, s)$ for candidate ranking and set the frame-level confidence $C_f(t, s) = P_{\text{frame}}(t, s)$ for Stage 3.

**Matched segment proposals and reference candidate set.** Let $\hat{s}(t) = \arg\max_s P_{\text{frame}}(t, s)$ be the top-1 matched

reference index, and let $A_{t,\hat{s}(t)}$ be its alignment logit. We form matched segment proposals by grouping consecutive $t$ that satisfy $|(\hat{s}(t+1) - \hat{s}(t)) - 1| \leq \delta$, where $\delta \geq 0$ is an integer tolerance for near-diagonal progression. We discard segments shorter than a minimum length, cap the segment length by $L_{\max}$, and optionally split segments at indices with $A_{t,\hat{s}(t)}$ below a threshold. For any uncovered query index $t$, we add a singleton segment $\{t\}$ to ensure full coverage.

For each query keyframe $t$, let $\mathscr{R}(t)$ be the set of reference indices contained in segment ranges covering $t$. By construction, $\mathscr{R}(t)$ is non-empty. We select one segment-consistent anchor

$$s_t^{\text{seg}} = \arg\max_{s \in \mathscr{R}(t)} P_{\text{frame}}(t, s), \tag{6}$$

and fill the remaining slots with globally top-ranked references under $P_{\text{frame}}(t, \cdot)$, with deduplication and $|\mathscr{S}_t| \leq |\mathscr{S}_t|_{\max}$.

### 4.3. Stage 2: Patch-level Correspondence and Change Feature Extraction

**Local patch correspondence.** For each $(t, s)$ with $s \in \mathscr{S}_t$, we compute dot-product local correlation volume between the query feature at $(x, y)$ and reference features at each offset in a $k \times k$ window, and apply a softmax over the $k^2$

*Table 3.* **Overall performance on the VSCD benchmark.** We report frame-wise F1 score (%) on the synthetic dataset and the real-world test set, together with results stratified by video length, graphic level, and the number of changed objects.

| Task | Method Model | Video length | | | Graphic level | | Object changes | | | Synthetic Overall | Captured by | | Real-world Overall |
|---|---|---|---|---|---|---|---|---|---|---|---|---|---|
| | | Low | Mid | High | Mid | High | Low | Mid | High | | Robot | Human | |
| AOD | TCF-LMO | 17.1 | 20.1 | 22.2 | 18.7 | 20.8 | 15.2 | 21.0 | 21.8 | 19.7 | 7.4 | 12.0 | 10.3 |
| | PBCD-MC | 24.6 | 28.0 | 26.5 | 26.3 | 27.4 | 23.4 | 28.0 | 27.8 | 26.8 | 16.8 | 15.7 | 16.1 |
| SCD | CSCDNet | 18.8 | 21.4 | 16.9 | 21.7 | 17.5 | 20.0 | 20.1 | 19.0 | 19.8 | 8.2 | 9.6 | 9.1 |
| | DR-TANet | 20.5 | 22.0 | 17.2 | 22.9 | 17.9 | 20.8 | 20.5 | 20.6 | 20.6 | 9.1 | 13.1 | 11.6 |
| | C-3PO | 22.7 | 26.1 | 20.8 | 30.1 | 16.9 | 18.8 | 25.1 | 27.8 | 24.1 | 8.0 | 13.9 | 11.7 |
| | ZSSCD | 0.1 | 0.5 | 0.1 | 0.5 | 0.1 | 0.2 | 0.4 | 0.2 | 0.3 | 2.1 | 0.0 | 0.8 |
| | GeSCF | 26.4 | 30.4 | 31.2 | 29.3 | 29.8 | 28.0 | 29.7 | 30.6 | 29.5 | 25.1 | 12.6 | 17.3 |
| VSCD | Ours | **38.1** | **36.9** | **33.9** | **40.7** | **31.7** | **32.1** | **37.7** | **39.0** | **36.6** | **32.0** | **21.5** | **25.4** |

offsets to obtain $P_{\text{patch}}^{(t,s)} \in \mathbb{R}^{k^2 \times H/p \times W/p}$, where $p$ denotes the ViT patch size.

**Differentiable warping.** We use the patch matching distribution to compute an expected displacement field on the patch grid. Let $\{\delta_i\}_{i=1}^{k^2}$ be the 2D offsets in the local window. We compute the expected displacement on the patch grid by

$$\Delta^{(t,s)}(x,y) = \sum_{i=1}^{k^2} P_{\text{patch},i}^{(t,s)}(x,y)\, \delta_i, \qquad (7)$$

and warp the reference feature map by bilinear sampling to obtain $E_{t,s}^{r(w)} \in \mathbb{R}^{D \times H/p \times W/p}$.

**Per-candidate change feature at low resolution.** For each $(t,s)$, we compute a compact change feature on the patch grid:

$$F_{t,s} = g_\phi\left(E_t^q, E_{t,s}^{r(w)}\right) \in \mathbb{R}^{C \times H/p \times W/p}, \qquad (8)$$

where $g_\phi$ is a lightweight convolutional head. It reduces channel dimensions of $E_t^q$ and $E_{t,s}^{r(w)}$, and fuses the reduced features with their cosine similarity map and difference to produce $F_{t,s}$.

### 4.4. Stage 3: Confidence-aware Feature Fusion and Query-guided Decoding

**Patch-level confidence.** We derive a patch-level confidence map from the patch matching distribution $P_{\text{patch}}^{(t,s)}$ produced in Stage 2, where $k$ is the local window size. At each spatial location $(x,y)$, we combine the peakiness of the distribution and its uncertainty:

$$p_{\max}^{(t,s)}(x,y) = \max_{i \in \{1,\dots,k^2\}} P_{\text{patch},i}^{(t,s)}(x,y), \qquad (9)$$

$$e^{(t,s)}(x,y) = -\frac{1}{\log(k^2)} \sum_{i=1}^{k^2} P_{\text{patch},i}^{(t,s)}(x,y) \log P_{\text{patch},i}^{(t,s)}(x,y), \qquad (10)$$

$$C_{sp}^{(t,s)}(x,y) = c_p\, p_{\max}^{(t,s)}(x,y) + c\left(1 - e^{(t,s)}(x,y)\right), \qquad (11)$$

where $C_{sp}^{(t,s)}(x,y)$ is the patch-level confidence for candidate pair $(t,s)$ at location $(x,y)$, and $c_p$ and $c$ are scalar weights.

**Confidence-aware feature fusion.** For each query keyframe $t$, we fuse $\{F_{t,s}\}_{s \in \mathscr{S}_t}$ with weights combining frame-level confidence $C_f(t,s) = P_{\text{frame}}(t,s)$ and patch-level confidence $C_{sp}^{(t,s)}$:

$$F_t(x,y) = \frac{\sum_{s \in \mathscr{S}_t} C_f(t,s) C_{sp}^{(t,s)}(x,y) F_{t,s}(x,y)}{\sum_{s \in \mathscr{S}_t} C_f(t,s) C_{sp}^{(t,s)}(x,y) + \varepsilon}, \qquad (12)$$

where, $F_t \in \mathbb{R}^{C \times H/p \times W/p}$ is the fused change feature for query frame $t$, and $\varepsilon$ is a small constant for numerical stability. This fusion is performed in feature space and yields a single fused representation per query frame, which enables decoding exactly once per query frame even when multiple reference candidates are used.

**Query-guided decoding.** Given the fused feature map $F_t$, we predict a full-resolution change mask logit map $\hat{Z}_t \in \mathbb{R}^{1 \times H \times W}$ using a lightweight decoder with progressive upsampling. To recover fine boundaries and high-frequency details that are not preserved at the patch-token resolution, we inject the query RGB frame into the decoder at two decoding resolutions. The change probability map is given by $\sigma(\hat{Z}_t)$, where $\sigma(\cdot)$ denotes the sigmoid function.

We train VSCDNet with change-mask supervision using BCE-with-logits and soft Dice losses; details are provided in Appendix B.

## 5. Experiments

### 5.1. Settings

**Baselines.** We compare VSCDNet against baselines from two related tasks: image-pair SCD and video-pair AOD. For SCD, we evaluate CSCDNet (Sakurada et al., 2020), DR-TANet (Chen et al., 2021b), and C-3PO (Wang et al., 2023), ZSSCD (Cho et al., 2025) and GeSCF (Kim & Kim,

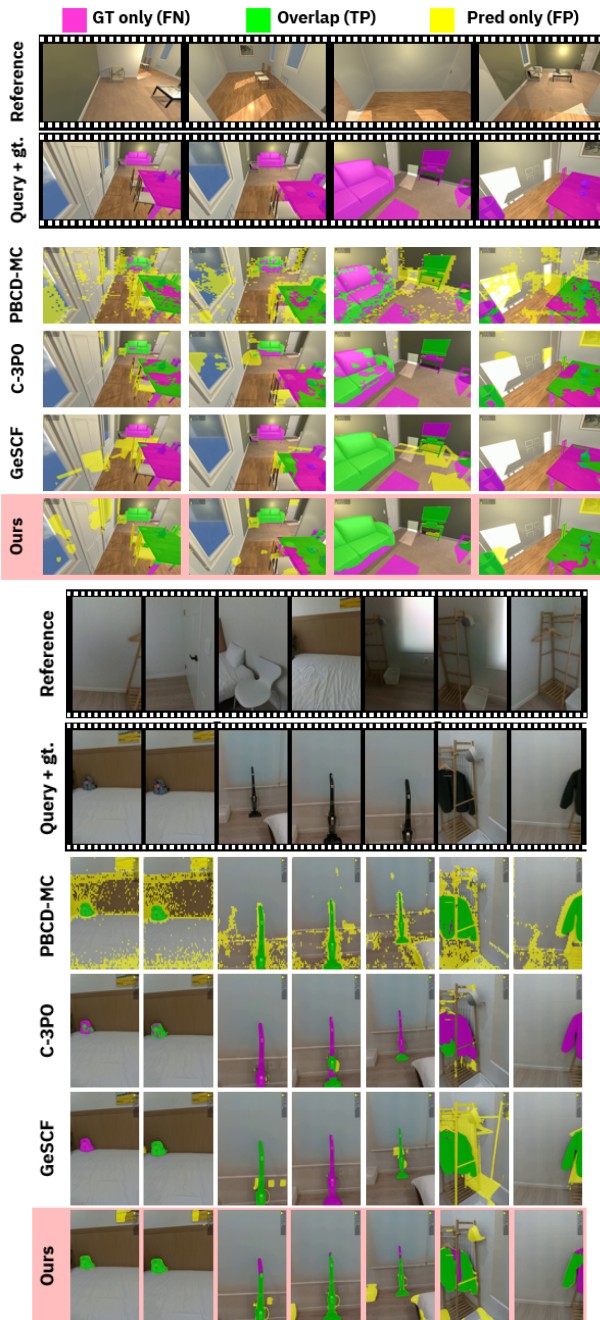

*Figure 3.* **Qualitative comparison results on VSCD dataset.** The top row shows results on the synthetic data, and the bottom row shows results on the real-world data.

2025). For AOD, we include TCF-LMO (Padilla et al., 2023) and PBCD-MC (Tavares et al., 2025). We do not include methods that require camera pose, geometric calibration, or explicit 3D/multi-view reconstruction as direct baselines, because such inputs and intermediate representations are unavailable under the VSCD protocol.

*Table 4.* **Module ablation study results.** We report the overall F1 score (%) on the synthetic test set.

| Variant | AT | $C_f$ | $C_{sp}$ | F1 (%) |
|---|---|---|---|---|
| Full (default) | O | O | O | **36.6** |
| w/o AT | X | O | O | 35.5 |
| w/o $C_{sp}$ | O | O | X | 35.9 |
| w/o $C_f$ | O | X | O | 35.1 |
| w/o $C_{sp}, C_f$ | O | X | X | 34.8 |

**Metrics.** We use pixel-wise F1 score as the primary metric, which is appropriate under severe class imbalance. All results are reported using Frame-wise F1, computed per frame and averaged over frames.

## 5.2. Comparative Studies

### 5.2.1. QUANTITATIVE COMPARISON

Table 3 reports frame-wise F1 on the synthetic benchmark and the real-world test set, with breakdowns by video length, graphic level, and the number of changed instances. VSCD-Net achieves the best overall performance on both domains (36.6 on synthetic and 25.4 on real-world). Importantly, the gains are not confined to a specific subset: VSCDNet remains top-1 across all slices and shows its clearest advantage as change density increases, with performance rising from 32.1 to 39.0 across low-to-high object-change settings. Moreover, VSCDNet sustains strong performance under long trajectories and high-quality rendering (33.9 and 31.7), which demonstrates that its improvements persist under substantial viewpoint drift and appearance variation rather than relying on a specific condition.

### 5.2.2. QUALITATIVE COMPARISON

Fig. 3 presents qualitative comparisons between our method and representative baselines from different change detection paradigms: PBCD-MC, C-3PO, and GeSCF.

PBCD-MC exhibits highly noisy change masks, particularly when the reference and query videos are captured from different viewpoints. As this method relies on constrained camera motion assumptions, its predictions degrade significantly under unconstrained trajectories. C-3PO produces relatively clean masks but often fails to fully cover changed object regions, resulting in fragmented or incomplete detections. This limitation stems from its reliance on single reference–query image pairs, which makes it difficult to resolve occlusions and viewpoint-induced misalignment. GeSCF, built upon a segmentation foundation model, effectively fills object regions when alignment is favorable. However, under strong misalignment, it frequently misclassifies large portions of the scene as changes, and it leads to many false positives.

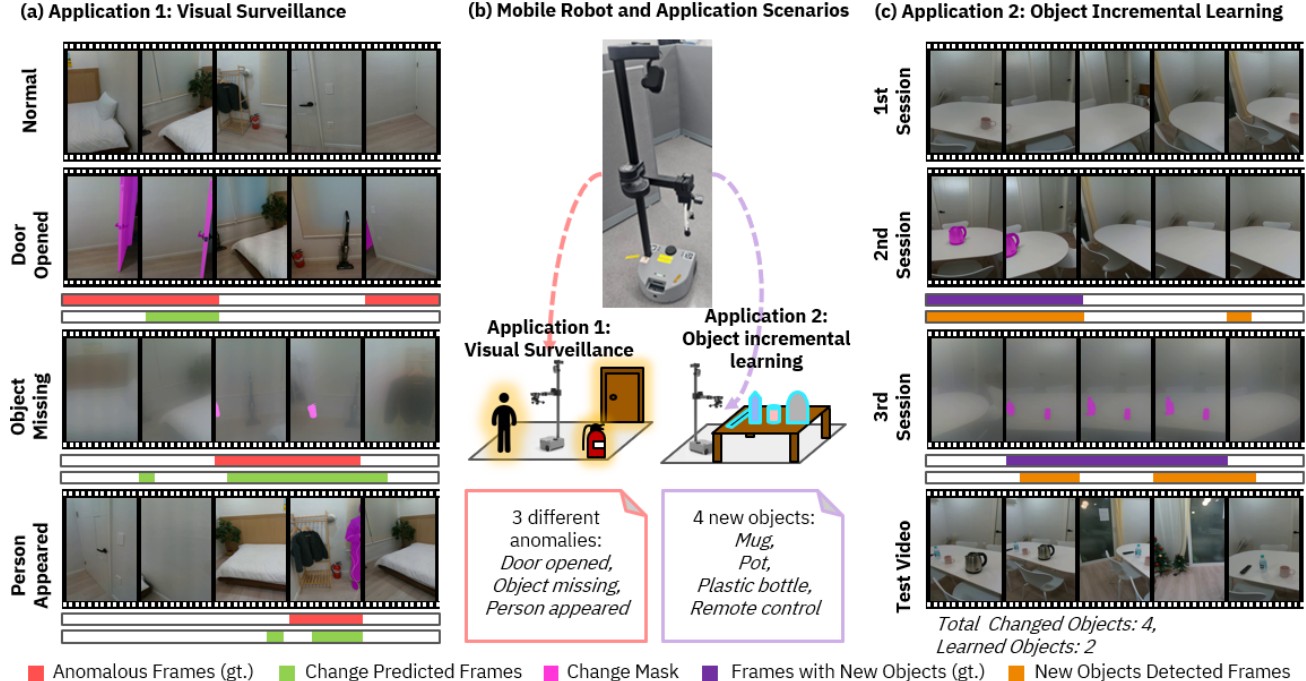

*Figure 4.* **Real-world robotic application of video scene change detection.** A mobile robot captures reference and query videos under unconstrained motion. VSCD supports (left) visual surveillance by localizing abnormal changes and (right) object incremental learning by highlighting newly introduced objects over time.

In contrast, our method produces stable and accurate change masks even under severe viewpoint differences. By leveraging sequence-level alignment, patch-level correspondence, and confidence-aware multi-reference fusion, our model remains robust to misalignment while preserving precise object-level change localization.

### 5.3. Ablation Studies

#### 5.3.1. MODULE ABLATION

Table 4 isolates three components that directly operationalize our core VSCD insight.

Removing the alignment token results in a moderate drop in performance, from 36.6 to 35.5. This indicates that a learned global frame representation improves frame-level matching under viewpoint variation, and it leads to higher-quality reference candidates and more reliable downstream fusion.

Global reliability matters more than local reliability, but both are complementary. Disabling $C_f$ produces the largest single drop, confirming that multi-reference reasoning becomes fragile if all candidates are treated equally: a few weakly matched frames can inject systematic noise into the fused representation. Disabling $C_{sp}$ yields a smaller but consistent degradation, indicating that correspondence ambiguity is often localized (e.g., repeated textures, thin structures, partial occlusions) and benefits from spatially varying trust.

When both confidence terms are removed, performance drops further, and it demonstrates that $C_f$ and $C_{sp}$ address different failure modes. $C_f$ decides which frames to trust, while $C_{sp}$ decides which pixels/patches within a trusted frame are geometrically reliable. Together they implement a principled trust pipeline that is essential in the VSCD regime where alignment is inherently uncertain.

## 6. Real-world Application

We deploy VSCDNet on a Stretch 3 mobile robot to validate VSCD beyond simulation under realistic sensing artifacts (Fig. 4). We demonstrate VSCDNet in two real-world applications: visual surveillance and object incremental learning. These applications use the binary mask as a reusable object-level change primitive: the same output is aggregated into anomaly scores for surveillance and used to restrict grounding/detection to changed regions for incremental learning. This design separates the hard VSCD problem of localizing persistent occupancy changes under unconstrained viewpoint mismatch from downstream semantic interpretation, such as assigning appearance, disappearance, or relocation labels.

For visual surveillance, we compare a single anomaly-free reference video against multiple query videos recorded at later times, and aggregate the predicted pixel-wise change

masks into frame-level anomaly scores to identify when abnormal changes occur (e.g., door state changes or object appearance/disappearance) without requiring trajectory replay. In our deployment, VSCDNet identifies frames containing the abnormal event with a hit rate of 62.1%. This result shows that our VSCD formulation supports temporal localization of persistent object-level changes from unaligned moving-camera videos.

For object incremental learning, we run VSCD between consecutive robot recordings and use the resulting change regions as proposals for newly introduced objects, which reduces the need for exhaustive manual annotation across the entire scene. In an evaluation in which 4 new objects are introduced over time, the system successfully classifies 2 of them in the final test run. This application indicates that VSCD can serve as an effective change-driven front-end for updating an object inventory under unconstrained robot motion. Implementation and data-collection details are provided in Appendix D.

## 7. Conclusion

We introduced Video-based Scene Change Detection (VSCD), where a reference and a query RGB video of the same environment are recorded at different times under unconstrained camera motion, without temporal synchronization or geometric calibration. We built a large-scale benchmark with over 1.1 million synthetic frames and a real-world test set to evaluate transfer beyond simulation. We proposed VSCDNet, a query-centric multi-reference model that performs frame-level alignment, patch-level correspondence, and confidence-weighted fusion to predict a high-resolution change mask once per query frame. VSCDNet outperforms strong image-based and video-based baselines on both synthetic and real-world evaluations. Further, we demonstrated the practical deployment of VSCDNet on a mobile robot. Future work includes expanding the real-world benchmark, domain adaptation, or few-shot real-world fine-tuning, outdoor and multimodal VSCD, and improved robustness under severe occlusion or large viewpoint gaps.

## Acknowledgements

This research was partly supported by the National Research Foundation of Korea (NRF) grant funded by the Korea government (MSIT) (No. NRF-2022R1C1C1009989); by the InnoCORE program of the Ministry of Science and ICT(GIST InnoCORE KH0860); by Institute of Information & communications Technology Planning & Evaluation (IITP) grant funded by the Korea government(MSIT) (No. RS-2022-II220926, Development of Self-directed Visual Intelligence Technology Based on Problem Hypothesis and Self-supervised Methods); and by the National Research Council of Science & Technology(NST) grant by the Korea government(MSIT) (No. GTL25041-000).

## Impact Statement

VSCD can support long-term robot monitoring by localizing persistent object-level changes from moving-camera videos. Because such monitoring may involve privacy-sensitive video data, deployment should require appropriate consent, access control, and data-retention practices. In safety-relevant use, false positives may trigger unnecessary inspection and false negatives may miss important changes; VSCD outputs should therefore be used as assistive signals rather than sole decision criteria.

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

# A. Dataset and Annotation Details

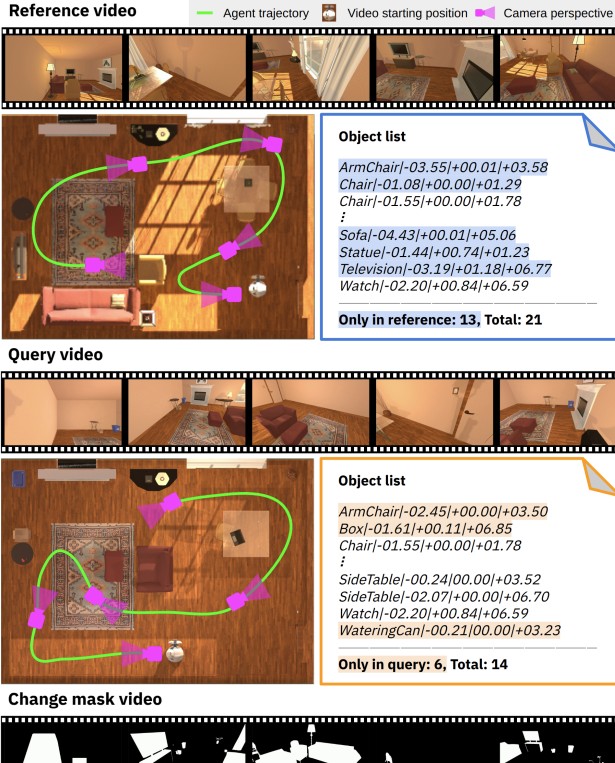

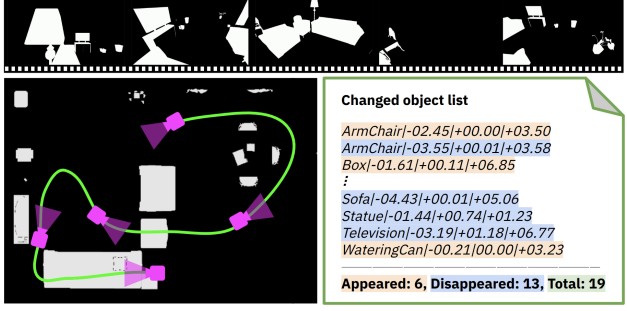

*Figure 5.* **Example of generating data in the AI2Thor simulator** (Kolve et al., 2017a). Reference and query videos are captured along different trajectories in the same environment with varying object configurations, and pixel-wise change masks are generated by rendering changed objects along the query trajectory.

## A.1. Synthetic Data Generation Details

### A.1.1. SIMULATORS AND RENDERING FIDELITY

We generate synthetic VSCD data using two simulation stacks: Unity-based AI2-THOR (Kolve et al., 2017a), a widely used simulator for embodied/robotics research, and Unreal Engine, which offers more photorealistic graphics. This design intentionally introduces rendering diversity: throughout the paper, Mid graphics correspond to Unity/AI2-THOR renders, while High graphics correspond to Unreal Engine renders (used in the graphics-level breakdown in Section 5.2).

### A.1.2. LAYOUTS, SCENES, AND DIRECTED SCENE PAIRS

Our benchmark comprises 218 pre-built indoor layouts (e.g., homes, offices, restaurants). For each layout, we create five scenes by varying the set and configuration of object instances while keeping the underlying layout geometry fixed. We then form VSCD examples as directed scene pairs ($s \rightarrow s'$) following a fixed cycle (e.g., $0 \rightarrow 1, 1 \rightarrow 2, \dots, 4 \rightarrow 0$), so that each scene acts as the query once. Object-level changes between $s$ and $s'$ include appearances, disappearances, and relocations; relocations are treated as changes under our formulation by modeling instances at different physical locations as distinct elements of the instance set.

### A.1.3. INDEPENDENT TRAJECTORIES AND CROSS-VIEW MISALIGNMENT

For each scene, we record an RGB video with a mobile agent traversing the environment along a manually designed trajectory. Crucially, reference and query videos are not required to share the same trajectory. By capturing different scenes with different paths and viewpoints, the resulting reference–query pair exhibits strong viewpoint variation, occlusion, and temporal asynchrony, matching the intended VSCD setting.

### A.1.4. PIXEL-WISE CHANGE MASK GENERATION BY QUERY-TRAJECTORY RENDERING

Pixel-wise supervision is generated at render time by leveraging simulator metadata. For a directed pair ($s \rightarrow s'$), let $O_r$ and $O_q$ denote the object-instance sets in the reference and query scenes, and let $O_\Delta = (O_q \setminus O_r) \cup (O_r \setminus O_q)$ denote the symmetric difference. To obtain a mask aligned to the query frames, we perform an additional render pass that replays the query camera trajectory and renders only instances in $O_\Delta$ as white while setting all other regions to black. This produces a binary mask sequence that is temporally aligned with the query video and reflects physical object-level changes rather than raw appearance differences. Fig.5 provides a concrete example in AI2-THOR, where the reference and query videos traverse the same environment along different trajectories, and query-aligned change masks are generated by replaying the query trajectory and rendering only instances in $O_\Delta$. The example also highlights the multi-change regime of VSCD (e.g., multiple appeared and disappeared instances in a single pair), which makes naive frame differencing unreliable under cross-view misalignment.

## A.2. Real-world VSCD Test Set and Annotation Pipeline

### A.2.1. DATA COLLECTION PROTOCOL

For each real-world example, we record a reference video $V_r$ and a query video $V_q$ of the same indoor space at two

different time points. The two recordings are intentionally captured along different paths and viewpoints, without assuming temporal synchronization or geometric calibration, to match the VSCD setting. Between the two recordings, the environment undergoes object-level changes (additions, removals, and relocations), while lighting and other photometric conditions may vary naturally.

### A.2.2. WHY PIXEL-WISE CHANGE ANNOTATION IS DIFFICULT IN VSCD

Obtaining pixel-wise change masks in the real world is challenging for two reasons that are intrinsic to VSCD. First, cross-view misalignment means that even unchanged regions may look very different across recordings. Therefore, naive frame differencing or image-pair annotation is unreliable. Second, disappeared objects provide no direct pixels in the query. Although the corresponding physical location may still be visible in $V_q$, the object instance itself is absent, and it makes it hard to localize the changed region in the query view without additional viewpoints or explicit 3D reconstruction. These difficulties highlight that VSCD is not only challenging for models but also for obtaining supervision in real deployments.

### A.2.3. AUXILIARY RECORDING FOR ANNOTATING DISAPPEARED OBJECTS

To mitigate the absence of direct visual evidence for disappeared objects, we capture an auxiliary RGB video $V_u$ for each pair, recorded only for annotation. In this auxiliary recording, all objects that appear in either $V_r$ or $V_q$ are physically present at once, and we traverse the scene with a trajectory similar to that of the query recording so that changed objects are observed under query-like viewpoints. This auxiliary video is never used as an input to any VSCD model and serves only to recover pixel-wise labels.

### A.2.4. WARP-AND-SEGMENT LABEL GENERATION

We generate per-frame change labels for the query video by combining (i) segmentation masks extracted directly from query frames (for appeared/visible changed objects) and (ii) segmentation masks extracted from auxiliary frames, warped into the query view (to recover disappeared-object regions).

Concretely, for each query frame $F_q^t$, we select an auxiliary frame (or a short local neighborhood of frames) that best matches the viewpoint of $F_q^t$ and estimate a warp that maps the auxiliary frame into the query frame coordinate system. Let $\mathscr{S}(\cdot)$ denote an off-the-shelf segmentation model used only for label generation; in our implementation, $\mathscr{S}$ is SAM2 (Ravi et al., 2024). We compute query masks $A^t = \mathscr{S}(F_q^t)$ and auxiliary masks $U^t = \mathscr{S}(F_u^{\pi(t)})$, and warp

the auxiliary masks into the query view to obtain $\widetilde{U}^t$.

We then combine masks to form the final change mask $M^t$ for each query frame:

$$M^t = M_{\text{app}}^t \cup M_{\text{dis}}^t,$$

where $M_{\text{app}}^t$ represents masks of appeared/relocated instances visible in the query, derived from $A^t$, and $M_{\text{dis}}^t$ represents regions corresponding to disappeared/relocated instances, derived from the warped auxiliary masks $\widetilde{U}^t$. Relocations are treated as changes by construction, contributing to both appeared and disappeared regions at different physical locations. This procedure produces approximate pixel-wise supervision that is aligned to the query frames and enables evaluation of sim-to-real generalization under realistic sensing noise. Real-world labels may contain boundary errors from SAM2 and view-warping artifacts for disappeared objects; since all methods are evaluated with the same labels, relative comparisons remain meaningful, while absolute real-world F1 should be interpreted conservatively.

## B. Implementation and Experimental Details

### B.1. Training and Implementation

**Keyframe sampling and preprocessing.** We train VSCD-Net with change-mask supervision on a fixed number of keyframes. For each reference–query pair, we uniformly sample $T_{\text{key}} = 32$ frames from both videos. When the original sequence is shorter than $T_{\text{key}}$, we allow repeated sampling so that each sample always contains exactly 32 frames per video. Each RGB frame is resized to $1024 \times 1024$ and normalized with ImageNet mean and standard deviation.

**Backbone and feature resolution.** We use a pretrained SAM ViT-B image encoder to extract patch tokens on a $64 \times 64$ grid ($D = 768$). From the patch tokens, we compute a compact per-frame descriptor by combining (i) average pooling over patch tokens and (ii) a learnable alignment token updated by a lightweight attention-based aggregation head (1 layer, 8 heads). All patch-level modules operate on the $64 \times 64$ token grid.

**Frame-level alignment and reference candidate construction.** Given query/reference frame descriptors, we build a cosine-similarity matrix and refine it with a shallow residual convolutional head. We obtain a soft matching distribution by applying a row-wise softmax with temperature $\tau_f = 0.5$. Matched segment proposals are constructed from the top-1 path using a near-diagonal tolerance $\delta = 2$, a minimum segment length of 2, and a maximum segment length $L_{\max} = 5$. For each query keyframe $t$, we select one segment-consistent anchor reference and fill the remaining slots with globally top-ranked references under the soft matching distribution,

using $K = 4$ and capping the final candidate set size to $|S_t| \leq 6$.

**Patch correspondence, fusion, and decoding.** For each candidate pair $(t, s)$, we compute local patch correspondence by evaluating dot-product correlation within a $k \times k$ window on the token grid (default $k = 5$), and apply softmax over the $k^2$ offsets to obtain a patch-matching distribution. We compute an expected displacement field and warp the reference feature map by differentiable bilinear sampling. A lightweight convolutional head then produces a per-candidate change feature at $64 \times 64$ resolution, and multiple candidate features are fused by a confidence-weighted average in feature space. Finally, we decode the fused feature once per query keyframe using a lightweight upsampling decoder ($64 \rightarrow 128 \rightarrow 256 \rightarrow 512 \rightarrow 1024$). To recover fine boundaries, we inject the query RGB frame into the decoder at the 256 and 1024 stages (16-channel projection), and output a full-resolution logit mask for each query frame.

**Loss and optimization.** We supervise change prediction using binary cross-entropy with logits and a soft Dice loss:

$$\mathscr{L}_{\text{mask}}^{(t)} = \mathscr{L}_{\text{BCEWithLogits}}(\hat{Z}_t, M_t) + \mathscr{L}_{\text{Dice}}(\hat{Z}_t, M_t), \quad (13)$$

and average $\mathscr{L}_{\text{mask}}^{(t)}$ over query keyframes in the batch. We optimize trainable parameters using AdamW while keeping the pretrained transformer image encoder frozen.

**B.2. Evaluation Protocol**

**Keyframe sampling.** For all methods, we uniformly sample a fixed number of keyframes from each reference and query video to control computational cost and ensure fair comparison. Unless otherwise stated, we use the same keyframe sampling strategy for all baselines and our method.

**Image-based SCD baselines.** Image-based SCD methods require a single reference image for each query frame. To adapt them to the VSCD setting, we perform explicit frame matching using ViT-SAM features (Kirillov et al., 2023). Specifically, we extract ViT-SAM embeddings for all reference and query keyframes, compute cosine similarity between them, and select the top-1 most similar reference frame for each query frame. The selected image pairs are then processed independently to produce pixel-wise change masks.

**Video-based AOD baselines.** Video-based abandoned object detection methods are applied directly to reference and query keyframe sequences following the original implementations. These methods rely on constrained camera motion and sequence-level comparison and do not require additional frame matching beyond their native pipelines.

*Table 5.* **Stratification criteria.** Difficulty factors used for the stratified breakdown in Table 3.

| Factor | Bin | Threshold |
|---|---|---|
| Video length | Low | < 900 frames |
| | Mid | 900–1500 frames |
| | High | > 1500 frames |
| Graphic level | Mid | Unity/AI2-THOR |
| | High | Unreal Engine |
| Object changes | Low | < 5 changed instances |
| | Mid | 5–15 changed instances |
| | High | > 15 changed instances |

*Table 6.* **Hyper-parameter ablation study.** Default configuration is $K = 4$, $|\mathscr{S}_t|_{\text{max}} = 6$, $k = 5$, $L_{\text{max}} = 5$, and $T_{\text{key}} = 32$. Each row changes one parameter while keeping the others fixed to the default.

| Parameter | Value | F1 (%) |
|---|---|---|
| Default | – | 36.6 |
| $K$ | 2 | 36.6 |
| $K$ | 6 | 36.6 |
| $|\mathscr{S}_t|_{\text{max}}$ | 4 | 36.2 |
| $|\mathscr{S}_t|_{\text{max}}$ | 8 | 36.4 |
| $k$ | 3 | 35.9 |
| $k$ | 7 | 37.2 |
| $L_{\text{max}}$ | 3 | 36.5 |
| $L_{\text{max}}$ | 7 | 36.5 |
| $T_{\text{key}}$ | 16 | 34.7 |
| $T_{\text{key}}$ | 64 | 37.7 |

**Difficulty factor stratification.** Table 5 summarizes the stratification criteria for difficulty factors used in the breakdown of Table 3. We bucket test pairs by (i) video length measured in the number of frames, (ii) graphics level determined by the simulation stack (Mid: Unity/AI2-THOR, High: Unreal Engine), and (iii) the number of changed object instances between the reference and query scenes (including appearances, disappearances, and relocations).

## C. Additional Experimental Results

### C.1. Hyper-parameter Ablation Study

We additionally evaluate sensitivity to key hyper-parameters in Table 6. We vary one parameter at a time while keeping the others fixed to the default configuration. Specifically, we consider $K$, the number of top reference frames retrieved per query frame from frame matching, $|\mathscr{S}_t|_{\text{max}}$, the cap on the final candidate set size, $k$, the local window size used for patch correspondence, $L_{\text{max}}$, the maximum length of matched segment proposals, and $T_{\text{key}}$, the number of sampled keyframes per video.

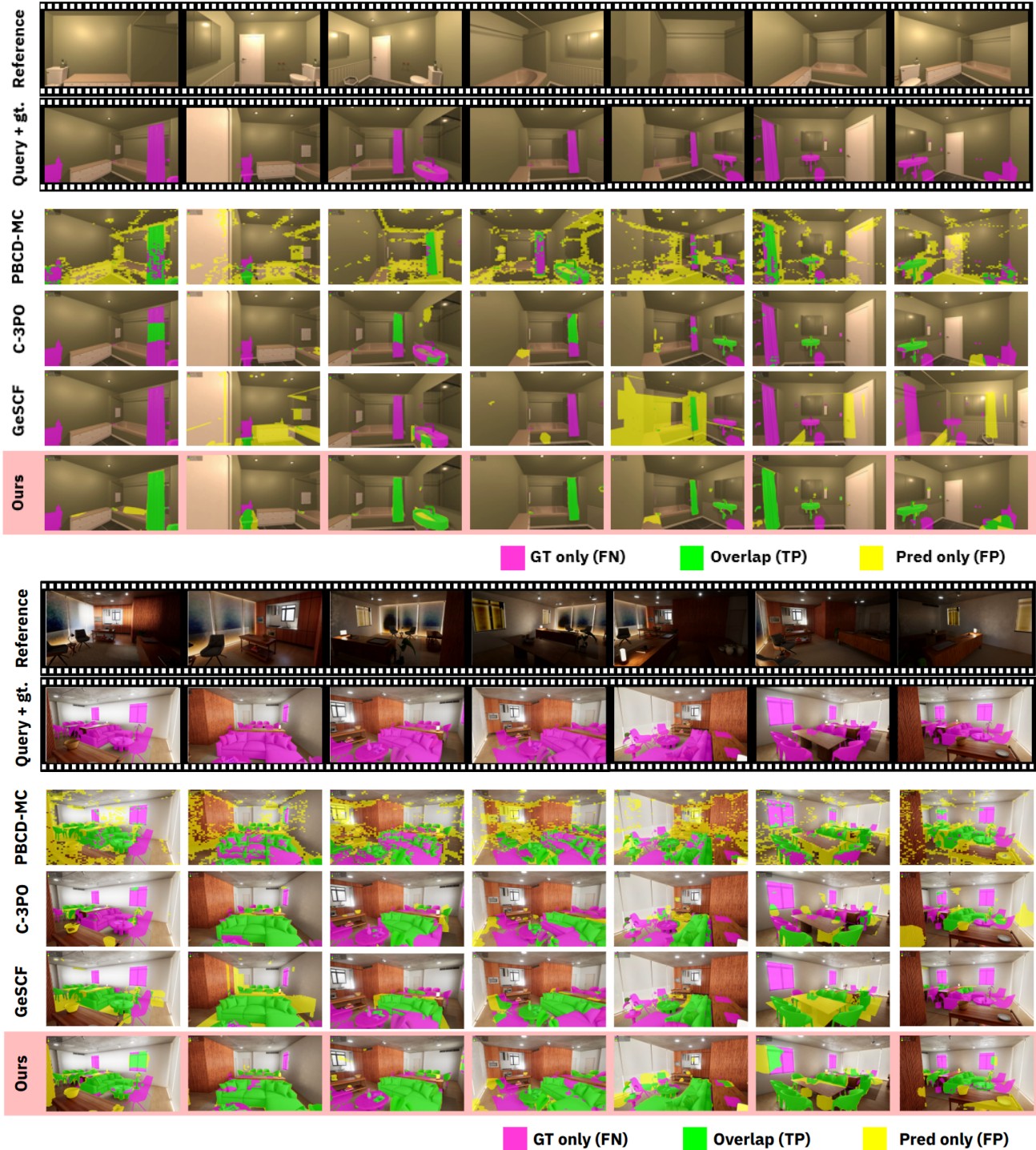

*Figure 6.* **Qualitative comparison results on synthetic VSCD dataset.** VSCDNet produces more coherent and complete change masks than representative baselines across both AI2-THOR (top; Mid graphics) and Unreal Engine (bottom; High graphics).

We first observe that varying $K$, the number of top reference frames retrieved per query frame from frame-level matching, has little effect on performance. Using $K = 2$ or $K = 6$ yields the same F1 score as the default setting. This indicates that the frame-level alignment stage already produces highly reli-

able reference candidates, such that increasing or decreasing the number of retrieved frames does not significantly affect the quality of the fused representation.

Similarly, changing the maximum candidate set size $|\mathscr{S}_t|_{\max}$

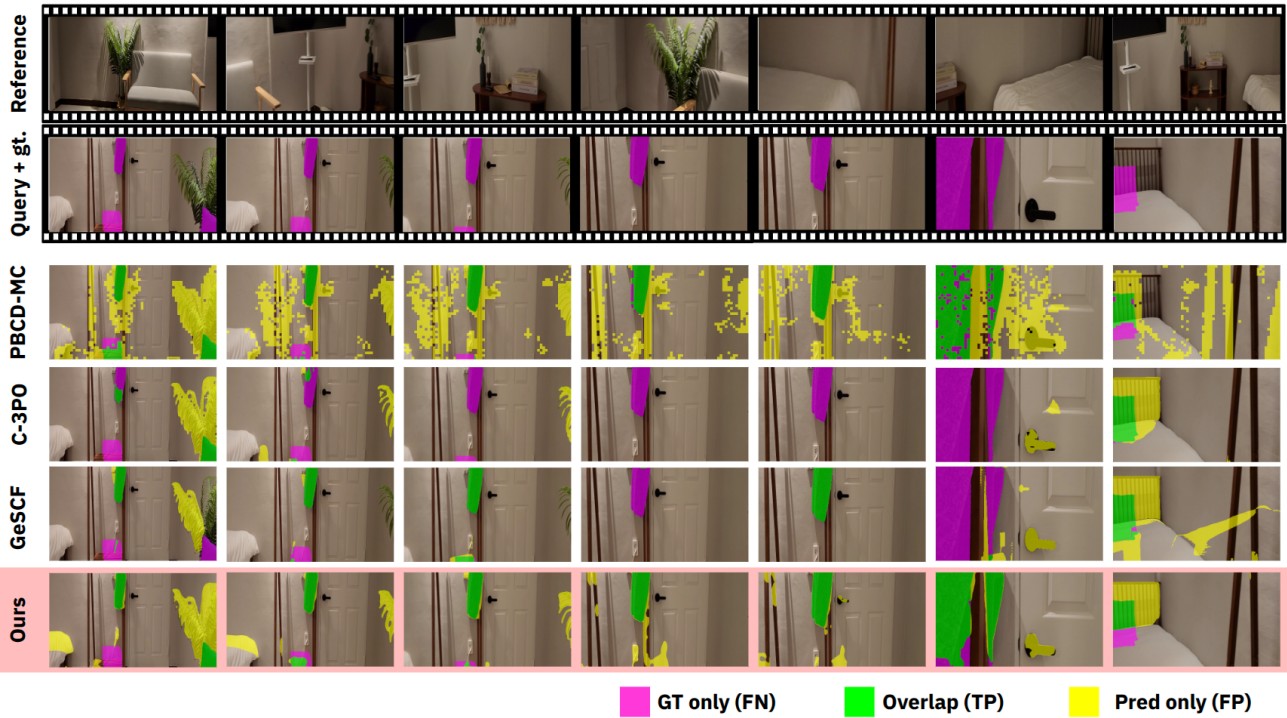

**Figure 7. Qualitative comparison results on real-world VSCD dataset.** Despite real sensing noise, illumination variation, and unconstrained camera motion, VSCDNet maintains coherent object-level change localization and reduces spurious detections compared to baselines.

results in only minor performance differences. This suggests that our candidate construction strategy is robust and does not rely on a large number of reference frames, as long as a small set of well-aligned candidates is available.

In contrast, the patch correspondence window size $k$ has a more noticeable impact. A larger window ($k = 7$) slightly improves performance, while a smaller window ($k = 3$) degrades it. This trend reflects a trade-off between capturing sufficient local displacement to compensate for viewpoint differences and maintaining precise spatial correspondence. We use $k = 5$ as the default because $k = 7$ improves real-world F1 only from 25.43 to 25.86, while increasing inference time from 3.44s to 3.62s and local patch-matching cost from 15.10 to 29.60 GMACs per video pair.

Varying $L_{max}$ between 3 and 7 results in nearly identical performance (36.5 F1 in both cases), comparable to the default setting. This indicates that VSCDNet is not sensitive to the exact temporal span of matched segments, as long as short spurious matches are filtered and overly long segments are avoided. In practice, a moderate segment length is sufficient to capture temporal consistency without introducing additional noise from loosely aligned frames.

Reducing $T_{key}$ to 16 leads to a noticeable drop in performance (34.7 F1), suggesting that too sparse temporal sam-

pling limits the ability to establish reliable cross-video associations. Increasing $T_{key}$ to 64 improves performance to 37.7 F1, indicating that denser temporal coverage provides additional matching evidence. However, the gain over the default setting remains moderate, and it reflects a trade-off between temporal resolution and computational cost. On the real-world set, using 8, 16, 24, and 32 sampled frames gives 0.87, 1.73, 2.62, and 3.44s forward time, with 3.19, 3.62, 4.05, and 4.48GB peak memory, respectively.

Overall, the limited sensitivity across hyper-parameters suggests that VSCDNet's performance is primarily driven by its core design—multi-reference alignment and confidence-aware fusion—rather than careful tuning of individual parameters.

### C.2. Alignment-quality Analysis

We analyze alignment quality using the $Top1 - Top2$ gap of $P_{frame}$, where smaller gaps indicate more ambiguous alignment. Table 7 clarifies the role of our design choices: ambiguous and medium cases benefit most from unweighted multi-reference reasoning, whereas clear cases benefit most from confidence-aware fusion. As alignment improves, both precision and recall rise, especially recall, suggesting that better alignment mainly recovers missed changed regions.

*Table 7.* **Alignment ambiguity analysis.** Query frames are grouped by the top-1/top-2 gap in $P_{\text{frame}}$; smaller gaps indicate more ambiguous alignment.

| Bin | Top-1 only | Multi-ref w/o conf | Full F1 | Precision | Recall |
|---|---|---|---|---|---|
| Ambiguous | 31.9 | 33.9 | 34.6 | 37.5 | 51.4 |
| Medium | 32.9 | 35.5 | 36.3 | 38.2 | 54.8 |
| Clear | 36.2 | 36.3 | 39.0 | 39.3 | 60.6 |

### C.3. Error Analysis

We also categorize frame-level errors: FP-heavy errors account for 53.9%, balanced errors for 27.7%, and FN-heavy errors for 18.4%. FPs mainly arise from small or near-zero-change frames with spurious unchanged regions, while FNs mainly reflect incomplete coverage of large changed objects.

### C.4. Synthetic-to-real Gap Analysis

To better understand the synthetic-to-real gap, we repeated the same failure analysis on both domains using change-size thresholds derived from the synthetic test split (Table 8). Real-world transfer is intrinsically difficult—due to motion blur, exposure changes, etc., yet VSCDNet still improves over the strongest baseline. The gap is not uniform: it is concentrated in small/medium changes, while large changes transfer much better. This indicates that sim-to-real degradation is driven mainly by fine/localized changes and boundary precision, not by a general failure of sequence-level alignment.

### C.5. Additional Qualitative Results

We provide additional qualitative comparisons to further validate the observations made in the main paper. Across a variety of environments, we consistently observe similar trends: video-based methods produce noisy predictions under viewpoint changes, image-based methods fail to fully capture object-level changes due to single-reference limitations, and zero-shot methods tend to over-predict changes under misalignment.

In contrast, our method maintains stable and coherent change masks across diverse scenes, even when reference and query videos exhibit large viewpoint differences or contain many simultaneous object-level changes. These additional results confirm that the qualitative advantages discussed in the main text generalize beyond the examples shown in Fig. 6, and Fig. 7.

## D. Real-world Application

This section demonstrates the deployment of the VSCD framework in real-world robotic applications, with a focus on two representative scenarios: visual surveillance and object incremental learning. All videos are collected using a Stretch 3 robot in indoor environments that include illumination changes, sensor noise, and viewpoint variation. Importantly, the reference and query videos are recorded under unconstrained robot motion without frame synchronization or pose information, reflecting realistic deployment conditions.

**Visual surveillance.** In the visual surveillance setting, we compare one anomaly-free reference video against three query videos that each contains a distinct abnormal event: (i) a door state change (closed → open), (ii) the appearance of a previously absent object, and (iii) the disappearance of an object present in the reference recording. The robot traverses the same environment across recordings but follows slightly different trajectories, producing strong cross-video misalignment and occlusion patterns. Given a reference–query pair, VSCD predicts a change mask sequence for the query video. For practical surveillance, we convert the predicted per-frame masks into frame-level anomaly scores (e.g., by aggregating mask confidence over pixels) and use these scores to identify frames and temporal intervals that contain abnormal changes. This application highlights a core capability of VSCD: reliable change detection under moving cameras without relying on fixed viewpoints or explicit frame correspondences, by aligning videos at the sequence level and consulting multiple reference observations for each query frame.

**Object incremental learning.** We further apply VSCD as a change-driven front-end for object incremental learning, where new objects are introduced over time and the robot updates its object inventory sequentially. The data consist of three training videos recorded in chronological order, followed by a test video that contains all objects introduced across training. The protocol proceeds as follows. First, in the initial training video, we obtain object candidates using an off-the-shelf grounding/detection module and initialize an object set by assigning each discovered instance a new class identity (treated as an incremental label). For the second and third training videos, we run VSCD between consecutive recordings, using the previous video as reference and the current video as query. VSCD identifies regions that differ across time; when these regions overlap with grounded object candidates, we treat them as newly introduced object instances and add them as new classes

*Table 8.* **Synthetic-to-real gap by change size.** We report precision, recall, and F1 after grouping frames by the visible size of changed regions, using thresholds derived from the synthetic test split.

| Change mask size | Synthetic Precision | Synthetic Recall | Synthetic F1 | Real-world Precision | Real-world Recall | Real-world F1 |
|---|---|---|---|---|---|---|
| Small ($<2.70\%$) | 18.9 | 70.5 | 21.4 | 14.5 | 73.1 | 13.3 |
| Medium ($2.70\%$–$10.82\%$) | 39.8 | 55.5 | 41.9 | 27.6 | 52.3 | 34.3 |
| Large ($\geq 10.82\%$) | 51.2 | 47.3 | 45.6 | 41.7 | 57.6 | 46.3 |

(or update the corresponding representation if an instance is re-observed). Finally, in the test video, we evaluate whether the system can correctly classify objects among all classes accumulated so far, using the learned incremental object set. This setting emphasizes the practical insight that semantic change detection can serve as a natural supervisory signal for incremental learning: by focusing computation and supervision on regions flagged as changed, the robot avoids exhaustive annotation of entire scenes, while sequence-level alignment reduces false discoveries caused purely by viewpoint or photometric variation rather than genuine object-level novelty.

