# OpenReview forum: "VSCD: Video-based Scene Change Detection in Unaligned Scenes"
_ICML.cc/2026/Conference — ICML 2026 regular_

### Official Review · Reviewer_3mkf · 2026-03-10

**Soundness:** 3
**Presentation:** 3
**Significance:** 3
**Originality:** 4
**Overall Recommendation:** 4
**Confidence:** 3

**Summary:**

This paper addresses the critical limitation of existing scene change detection methods that fail to handle unconstrained camera motion, severe viewpoint misalignment, and multiple simultaneous object changes—key challenges for long-term autonomous agent operation. It formulates a novel task: Video-based Scene Change Detection (VSCD), which aims to predict pixel-wise object-level change masks for query video frames from unaligned reference and query RGB videos of the same indoor environment captured at different times (excluding illumination/shadow variations as non-meaningful changes). To support VSCD research, the authors construct a large-scale benchmark with 1.13M synthetic frames (218 environments) and an 17k real-world frame test set (8 environments), both with pixel-accurate change annotations. The core contribution is VSCDNet, a query-centric multi-reference model with a three-stage architecture: frame-level alignment for reference candidate selection (without trajectory synchronization), patch-level correspondence for geometric compensation (without pose supervision), and confidence-weighted feature fusion plus query-guided high-resolution decoding. Extensive experiments show VSCDNet outperforms state-of-the-art image-based SCD and video-based AOD baselines on both synthetic and real-world datasets, and real-world deployment on a Stretch 3 mobile robot validates its utility for visual surveillance and object incremental learning.

**Compliance With Llm Reviewing Policy:**

Affirmed.

**Final Justification:**

My concerns have been addressed, so I keep my ratings.

**Key Questions For Authors:**

1.	For the patch correspondence window size k, your hyperparameter study shows k=7 yields a slightly higher F1 (37.2%) than the default k=5(36.6%). Why did you select k=5 as the default instead of k=7—was the choice based on a trade-off between computational cost and performance? If so, could you provide quantitative results on the computational overhead (e.g., inference time, memory usage) for different k values?
2.	Your model currently focuses on indoor environments—have you tested or explored the generalization of VSCDNet to outdoor environments? If outdoor testing is not yet performed, what core challenges do you anticipate for adapting VSCD to outdoor scenes (e.g., dynamic background, larger viewpoint variation, weather effects), and what modifications to the model/dataset would be needed?
3.	The VSCD benchmark currently includes 8 real-world environments—do you plan to expand the real-world dataset to include more diverse indoor scenes (e.g., large open spaces, cluttered industrial environments) and capture modalities (e.g., different camera resolutions, depth cameras)?

**Limitations:**

1.	Discuss the generalization ability of non-RGB modalities: current models only use RGB videos - please elaborate on the limitations of change detection using only RGB input (such as poor performance under low-light conditions) and the potential for expansion to multi-modal input (such as depth and thermal imaging cameras).
2.	Labeling limitations: The real-world labeling process relies on auxiliary recording and the pre-trained segmentation model (SAM2), which introduces slight labeling noise. Please discuss the impact of this noise on model evaluation, as well as potential strategies for achieving more accurate real-world labeling.

**Strengths And Weaknesses:**

Strength:
1. Novel Task and Benchmark: The work introduces the first VSCD task and a dedicated large-scale benchmark for unaligned videos with unconstrained camera motion, setting a new research direction with high significance for embodied AI and computer vision.
2. Rigorous Validation: The study is technically sound with well-designed experiments, including stratified comparisons, ablation studies, and real-world robotic deployment tests that effectively support core claims.

Weaknesses:
1. Real-world performance is still moderate (25.4% F1) with obvious gaps from synthetic results, and the failure modes of real-world applications lack in-depth analysis.
The model’s scalability to longer video trajectories is not fully verified, and computational overhead trade-offs for key modules (e.g., larger patch windows) are not quantitatively presented.
2. The research is limited to indoor RGB video scenarios, with no exploration of generalization to outdoor or multi-modal (depth/thermal) inputs.

---

> ### Author Rebuttal · Authors · 2026-03-30
>
> We thank the reviewer for recognizing the originality of our work as excellent. We address each question in detail.
>
> Table B. Failure modes by change size on synthetic vs. real-world (%).
>
> | Change mask size | Synthetic Precision | Synthetic Recall | Synthetic F1 | Real-world Precision | Real-world Recall | Real-world F1 |
> |---|:---:|:---:|:---:|:---:|:---:|:---:|
> | Small (<2.70%) | 18.9 | 70.5 | 21.4 | 14.5 | 73.1 | 13.3 |
> | Medium (2.70%–10.82%) | 39.8 | 55.5 | 41.9 | 27.6 | 52.3 | 34.3 |
> | Large (≥10.82%) | 51.2 | 47.3 | 45.6 | 41.7 | 57.6 | 46.3 |
>
> 1. **On the synthetic-to-real gap.**
> To better understand the synthetic-to-real gap, we repeated the same failure analysis on both domains using change-size thresholds derived from the synthetic test split (Table B). Real-world transfer is intrinsically difficult—due to motion blur, exposure changes, etc., yet VSCDNet still improves over the strongest baseline by +8.1 F1 (25.4 vs. 17.3). The gap is not uniform: it is concentrated in small/medium changes, while large changes transfer much better. This indicates that sim-to-real degradation is driven mainly by fine/localized changes and boundary precision, not by a general failure of sequence-level alignment.
> 2. **Scalability to longer trajectories.**
> The paper reports a breakdown by video length in Table 3, where VSCDNet achieves 38.1 / 36.9 / 33.9 F1 on low/mid/high-length (<30s / 30–50s / ≥50s) synthetic pairs. Performance decreases moderately as videos become longer, but remains top-1 in every bin. As an additional scalability check, we varied the real-world sampled frame budget at test time from 8 → 16 → 24 → 32. This increased average forward time from 0.87s → 1.73s → 2.62s → 3.44s and peak memory from 3.19GB → 3.62GB → 4.05GB → 4.48GB, showing predictable scaling with larger test-time budgets.
> 3. **Computational trade-off for larger patch windows.**
> On the real-world set, $k=7$ improves F1 only slightly (25.43→25.86, +0.43) while increasing end-to-end inference time from 3.44 to 3.62 s per video pair (+5.35%). Inside local patch matching, the k-dependent computation nearly doubles (15.10→29.60 GMACs per video pair; 0.472→0.925 per query frame), consistent with the expected $k^2$ growth. Thus, $k=5$ is the more practical default.
> 4. **Outdoor generalization/beyond-RGB**
> We view outdoor VSCD as a substantially harder regime, in which persistent object changes must be separated from transient dynamics under greater viewpoint/scale variation and weather/illumination changes. Our contribution here is to establish the indoor RGB video-to-video setting first. Moreover, the formulation itself is not tied to RGB; RGB-only is more vulnerable under low-light, low-texture, or transparent-surface conditions, where depth or thermal could be incorporated into the same alignment/correspondence/fusion pipeline with minimal extension.
> 5. **Real-world dataset expansion.**
> Yes—expanding the real-world benchmark is important future work. We view the current real-world set as a first transfer testbed rather than the final scale of the problem. Increasing scene diversity (e.g., larger open spaces, cluttered workspaces, different cameras/resolutions, or depth-enabled capture) would improve evaluation breadth and help better separate viewpoint difficulty from domain shift.
> 6. **Annotation noise and impact.**
> The real-world labels are approximate by construction, derived from auxiliary recording, warping, and SAM2 (used only for annotation). The two main noise sources are: (i) segmentation imprecision at object boundaries and thin structures, and (ii) warping artifacts for disappeared objects. Crucially, the noise is method-agnostic, so relative rankings remain reliable, and reported F1 scores are, if anything, conservative for all methods uniformly. Potential improvements include multi-round human refinement of SAM2 boundaries, multi-view consensus from multiple auxiliary frames, and 3D-assisted projection to ensure geometrically consistent labels.

---

> > ### Author Rebuttal · Reviewer_3mkf · 2026-04-03
> >
> > Thank you for your rebuttal. My concerns have been addressed, and I will keep my ratings.

---

> > > ### Author Response · Authors · 2026-04-05
> > >
> > > Thank you for your thoughtful feedback and for confirming that your concerns have been addressed. We truly appreciate your time and careful review of our work.

---

### Official Review · Reviewer_MHg7 · 2026-03-11

**Soundness:** 2
**Presentation:** 3
**Significance:** 3
**Originality:** 2
**Overall Recommendation:** 4
**Confidence:** 3

**Summary:**

This paper studies video-based scene change detection from a reference video and a query video captured at different times under viewpoint and trajectory mismatch, and predicts a binary change mask for each query frame. The paper also introduces a large synthetic benchmark plus a smaller real-world test set, and proposes a three-stage model with frame matching, patch correspondence, and confidence-weighted fusion. Overall, I think the benchmark is a useful contribution, but I am less convinced by the novelty claims, the fairness of the baseline set, and the strength of the evidence for real-world applicability.

**Compliance With Llm Reviewing Policy:**

Affirmed.

**Final Justification:**

Thank you for your rebuttal. The authors have fully addressed my concerns. Especially in terms of practical value. So I raised my score to 4

**Key Questions For Authors:**

1. The paper’s novelty claim and empirical positioning would benefit from clearer scoping. Could the authors clarify the intended scope of the “first VSCD task” claim, and discuss more explicitly how this work differs from recent multi-view / unaligned change detection methods such as 3DGS-CD [1] and Multi-View Pose-Agnostic Change Localization with Zero Labels [2]? Relatedly, can the authors comment on why such closer baselines were not included in the empirical comparison?
2. The paper emphasizes robustness to viewpoint mismatch, occlusion, and misalignment, but the evidence is mostly indirect. Can the authors provide a more controlled analysis of misalignment severity, alignment quality, and representative failure cases? In particular, some qualitative examples (see Fig.2) appear to contain clear false positives (e.g., unchanged window regions being highlighted as changes), but the paper does not analyze why these errors occur or how common they are.
3. In the real-world demos, how much of the final result comes from VSCDNet itself, and how much comes from the additional downstream processing? In particular, do these demos really test more complex multi-object changes, or mostly simpler single-object cases?
4. Could the authors clarify whether the object configuration is assumed to remain fixed within each individual reference/query video, with changes occurring only between the two recordings? Based on Appendix A, the data generation seems to treat each video as a multi-view observation of a single static scene state, rather than a scene that evolves during recording. If so, would it be more accurate to frame the main challenge as multi-view geometry and cross-view matching, rather than temporal continuity within a video?


[1] Lu Z, Ye J, Leonard J. 3dgs-cd: 3d gaussian splatting-based change detection for physical object rearrangement[J]. IEEE Robotics and Automation Letters, 2025, 10(3): 2662-2669.

[2] Galappaththige C J, Lai J, Windrim L, et al. Multi-view pose-agnostic change localization with zero labels[C]//Proceedings of the Computer Vision and Pattern Recognition Conference. 2025: 11600-11610.

**Limitations:**

No. The paper does not provide a substantial discussion of limitations or potential negative societal impact. It only briefly mentions future work on longer trajectories and improved geometric robustness. A stronger discussion would be helpful, especially around the reliance on synthetic training, the limited real-world evaluation scale, the fact that the method predicts only binary change masks, the use of additional downstream processing in the robot demos, and failure modes under severe misalignment or occlusion. On societal impact, the paper should also discuss surveillance/privacy concerns and the consequences of false positives or missed detections in real deployments.

**Strengths And Weaknesses:**

Strengths：
1. The dataset contribution is meaningful. The paper targets an important embodied perception setting that is not well covered by standard image-pair change detection benchmarks, and the scale of the synthetic data is substantial.
2. The paper is well organized, and the proposed pipeline is easy to follow. The ablation study also gives some evidence that the main components are useful.

Weaknesses：
1. The novelty claim feels too broad. The paper presents VSCD as the first task of this kind, but several recent works already study closely related settings with unaligned or multi-view change detection, including 3DGS-CD[1] and Multi-View Pose-Agnostic Change Localization with Zero Labels[2]. These papers are not identical to the exact formulation here, but they make the “first” claim hard to support as written. The contribution seems better framed as a specific RGB-video benchmark, not the first study of the broader problem.
2. The baseline set is not fully fair for the claimed setting, so the empirical SOTA claim is not yet fully convincing.Most baselines are drawn from image-pair SCD or video-pair AOD, whereas the paper’s own setting emphasizes multi-view, unaligned, object-level change detection with multi-reference reasoning. In that sense, the paper convincingly shows superiority over the chosen binary-mask F1 baseline set, but it does not fully establish state of the art over the broader class of more closely related methods, especially methods that explicitly reason over multi-view or 3D structure. This is particularly relevant because the paper’s task assumptions are closer to those methods than to traditional pairwise SCD baselines.
3. The practical value is still somewhat limited. The model only predicts a binary change mask, without distinguishing appearance, disappearance, or relocation. In the robot demos, additional post-processing is needed: surveillance uses frame-level anomaly scoring from the masks, and incremental learning further relies on external grounding/detection modules. As a result, the demos support the method as a useful front-end, but not yet as a full semantic change understanding system, especially for more complex multi-object real-world scenarios.

[1] Lu Z, Ye J, Leonard J. 3dgs-cd: 3d gaussian splatting-based change detection for physical object rearrangement[J]. IEEE Robotics and Automation Letters, 2025, 10(3): 2662-2669.

[2] Galappaththige C J, Lai J, Windrim L, et al. Multi-view pose-agnostic change localization with zero labels[C]//Proceedings of the Computer Vision and Pattern Recognition Conference. 2025: 11600-11610.

---

> ### Author Rebuttal · Authors · 2026-03-30
>
> We thank the reviewer for the thoughtful and constructive feedback. We address each concern below.
> 1. **Novelty and scope.**
> We agree that precise scoping is important. Both (Lu et al., 2025) and (Galappaththige et al., 2025) require **known camera intrinsics (calibration) and camera poses (motion parameters)** and reconstruct 3D Gaussian Splatting models for inference. In contrast, VSCD assumes **no pose information, no calibration, and no 3D reconstruction**.
> We acknowledge that this broader area is well-studied, and do not claim to be the first to address change detection with any viewpoint variation. Our “first” claim is scoped to the VSCD formulation in Sec. 3.1: pixel-wise object-level change detection from two unaligned RGB videos under unconstrained motion, without pose, calibration, or synchronization.
> 2. **Baselines and empirical positioning.**
> We do not claim superiority over all multi-view or 3D-reconstruction methods. Such methods require additional structure or a different inference pipeline (e.g., calibration, pose, explicit 3D modeling)—they are not like-for-like baselines under our formulation. Under the VSCD benchmark and evaluation protocol, our model outperforms strong 2D image-based SCD and video-based AOD baselines that produce per-frame change masks.
> 3. **Practical value.**
> We respectfully argue that the binary mask formulation is the appropriate foundational step. The binary mask is a **general-purpose primitive** that makes VSCDNet a reusable front-end for embodied AI tasks. Our two robot demonstrations illustrate this: the same binary mask supports temporal anomaly localization and change-driven object discovery, each with minimal post-processing—checking for changes exceeding a threshold confidence and verifying objectness of change masks with a detector, respectively—while handling *multiple-objects*. Second, extending from binary to semantic change types is straightforward: appearance and disappearance follow asymmetric scene occupancy, and relocation is a disappearance at one location and an appearance at another. Third, binary change detection itself remains an **unsolved problem** in the VSCD setting: even the strongest baseline achieves an F1 of 29.5 (Table 3).
>
> Table A. Controlled alignment-quality analysis by ambiguity bin (F1, %).
>
> | Bin |  Top-1 only |  Multi-ref w/o conf |  Full F1 | Precision | Recall |
> |---|:---:|:---:|:---:|:---:|:---:|
> | Ambiguous | 31.9 | 33.9 | 34.6 | 37.5 | 51.4 |
> | Medium | 32.9 | 35.5 | 36.3 | 38.2 | 54.8 |
> | Clear | 36.2 | 36.3 | 39.0 | 39.3 | 60.6 |
>
> 4. **Alignment quality.**
> We analyze alignment quality using the $\text{Top}1–\text{Top}2$ gap of $P_\text{frame}$, where smaller gaps indicate more ambiguous alignment. Table A clarifies the role of our design choices: ambiguous and medium cases benefit most from unweighted multi-reference reasoning (+2.0/+2.6 F1 over Top-1), whereas clear cases benefit most from confidence-aware fusion (+2.7 F1 over unweighted multi-ref). As alignment improves, both precision and recall rise, especially recall (51.4→60.6 vs. 37.5→39.3), suggesting that better alignment mainly recovers missed changed regions.
> 5. **Representative failure cases.**
> We analyze frame-level errors. FP-heavy errors are the most common (53.9%), followed by balanced (27.7%) and FN-heavy (18.4%). The main failure modes are: (i) FP-heavy over-segmentation on small/near-zero-change frames, including spurious unchanged regions, and (ii) FN-heavy incomplete coverage on large changes.
> 6. **Task assumption.**
> Within each reference/query video, the scene state is fixed; changes occur between the two recordings. While cross-view matching is a central challenge of VSCD, we respectfully point out that temporal continuity is a video-specific structural prior that makes raw-video cross-view matching feasible without explicit 3D reconstruction. Stage 1 uses temporal ordering to restrict each query frame to a short, sequence-consistent candidate rather than the full reference video—turning an intractable all-pairs problem into efficient candidate selection. Thus, the most precise framing is VSCD as a query-centric video-to-video change detection problem whose core challenge is cross-view reasoning under severe misalignment, while temporal continuity within each recording provides the key cue. This differentiates VSCD from recent multi-view/3DGS-based methods, which aggregate observations into explicit 3D scene representations, whereas VSCD predicts a 2D change mask per query frame directly from raw, unsynchronized RGB videos.
> 7. **Limitations/societal impact.**
> Key limitations are reliance on synthetic training and the small scale of real-world evaluation. In deployment, false positives can trigger unnecessary follow-up actions, and false negatives can miss meaningful changes. Because surveillance is one application domain, privacy-sensitive use and access control are important.

---

> > ### Author Rebuttal · Reviewer_MHg7 · 2026-04-03
> >
> > Thank you for your rebuttal. The authors have fully addressed my concerns. Especially in terms of practical value.

---

> > > ### Author Response · Authors · 2026-04-05
> > >
> > > We are grateful for your encouraging follow-up and for recognizing that your concerns have been fully considered. We especially appreciate your positive assessment of the practical value of our work, as well as your updated evaluation.

---

### Official Review · Reviewer_hGdY · 2026-03-12

**Soundness:** 3
**Presentation:** 3
**Significance:** 3
**Originality:** 3
**Overall Recommendation:** 4
**Confidence:** 4

**Summary:**

This paper introduces Video-based Scene Change Detection (VSCD), a new task and benchmark for detecting pixel-wise object-level changes between two unaligned indoor videos recorded at different times. The authors present VSCDNet, a query-centric multi-reference model that uses temporal alignment, patch-level correspondence, and confidence-based fusion to produce accurate change masks. Extensive experiments show that VSCDNet outperforms both image- and video-based baselines on a large synthetic dataset and real-world tests. The method is also demonstrated in practical robot applications such as visual surveillance and incremental object learning.

**Compliance With Llm Reviewing Policy:**

Affirmed.

**Final Justification:**

The paper is technically sound, clearly written, and supported by strong experiments, including ablations and sim-to-real evaluation.
The rebuttal resolved my concern.

**Key Questions For Authors:**

The benchmark and method rely on large-scale synthetic data for training, with real-world sets used only for testing. Can the authors discuss the potential for domain adaptation or sim-to-real transfer strategies? Evidence or experiments addressing model robustness and fine-tuning on real-world data could increase confidence in its generalization and practical utility.

**Limitations:**

yes

**Strengths And Weaknesses:**

Strengths:
1. The method is well motivated by the actual challenges in VSCD. It directly addresses temporal misalignment, viewpoint differences, and noisy references through a staged design, which makes the overall approach feel grounded in the problem rather than arbitrary.
2. The experimental section is comprehensive, including systematic ablation studies, clear evaluations of both synthetic and real-world datasets, and a variety of parameter analyses.
3. The zero-shot sim-to-real setting is a strong point. Training only on synthetic data while still achieving good performance on real-world videos suggests solid generalization, and the frozen backbone with lightweight modules seems like a reasonable design choice.


Limitation:
Hope you can further explain why the structure and design of VSCD can accurately predict a pixel-wise change mask for each query frame.

---

> ### Author Rebuttal · Authors · 2026-03-30
>
> We thank the reviewer for recognizing the strength of our experiment. We address the raised points below.
> 1. **The effectiveness of the design.**
> VSCD has three characteristic failure modes: (i) naive frame pairing often compares inconsistent observations since query and reference videos are temporally unsynchronized; (ii) even after selecting the best-matching reference frame, severe viewpoint differences cause spatial misalignment at the pixel level; and (iii) consulting multiple reference frames is essential due to occlusion or limited overlap while not all retrieved frames or local matches are reliable—propagating noise. VSCDNet is designed to address these issues explicitly rather than with a single monolithic encoder. Stage 1 retrieves temporally consistent reference frames from unaligned videos, Stage 2 establishes patch-level correspondences to handle residual viewpoint differences without pose supervision, and Stage 3 uses confidence-aware fusion that suppresses unreliable global/local matches. The ablation confirms the effectiveness of each stage: removing temporal alignment or confidence consistently degrades F1, with the largest drop when frame-level confidence is removed (36.6→35.1), and removing confidence terms further reduces performance to 34.8. These results support that the gains come from matching the architecture to the specific failure modes of VSCD.
> 2. **Sim-to-real transfer.**
> Thank you for the suggestion. Our current protocol is intentionally zero-shot sim-to-real: the real-world split is reserved strictly for evaluation—no real-world fine-tuning is used. We view this as a strength of the benchmark, since it evaluates whether a method trained only on synthetic data can transfer to unconstrained real videos. Under this setting, VSCDNet achieves the best real-world performance (25.4 overall F1), outperforming the baselines on both robot-captured and handheld videos (32.0 and 21.5)—supporting its robustness and practical promise. We believe this transfer benefits from the frozen backbone, whose domain-general features anchor the learnable lightweight modules to acquire task-specific reasoning without overfitting. Further, we agree that domain adaptation and few-shot real-world fine-tuning are natural next steps.

---

> > ### Author Rebuttal · Reviewer_hGdY · 2026-04-03
> >
> > Thank you for the rebuttal.
> > The authors have adequately addressed my concerns.

---

> > > ### Author Response · Authors · 2026-04-05
> > >
> > > We sincerely appreciate your follow-up and your explicit acknowledgement that your concerns have been fully resolved. Thank you for your time and thoughtful evaluation of our work.

---

### Decision · Program_Chairs · 2026-04-30

**Decision:**

Accept (regular)

**Comment:**

The rebuttal has addressed all reviewers' concerns raised during the first round review process.  There is no further comment from the SAC/AC.